

# Synthetic dataset generation for object-to-model deep learning in industrial applications

Matthew Z. Wong, Kiyohito Kunii, Max Baylis, Wai Hong Ong, Pavel Kroupa and Swen Koller

Department of Computing, Imperial College London, London, UK

## ABSTRACT

The availability of large image data sets has been a crucial factor in the success of deep learning-based classification and detection methods. Yet, while data sets for everyday objects are widely available, data for specific industrial use-cases (e.g., identifying packaged products in a warehouse) remains scarce. In such cases, the data sets have to be created from scratch, placing a crucial bottleneck on the deployment of deep learning techniques in industrial applications. We present work carried out in collaboration with a leading UK online supermarket, with the aim of creating a computer vision system capable of detecting and identifying unique supermarket products in a warehouse setting. To this end, we demonstrate a framework for using data synthesis to create an end-to-end deep learning pipeline, beginning with real-world objects and culminating in a trained model. Our method is based on the generation of a synthetic dataset from 3D models obtained by applying photogrammetry techniques to real-world objects. Using 100K synthetic images for 10 classes, an InceptionV3 convolutional neural network was trained, which achieved accuracy of 96% on a separately acquired test set of real supermarket product images. The image generation process supports automatic pixel annotation. This eliminates the prohibitively expensive manual annotation typically required for detection tasks. Based on this readily available data, a one-stage RetinaNet detector was trained on the synthetic, annotated images to produce a detector that can accurately localize and classify the specimen products in real-time.

# INTRODUCTION

In this paper, we present a framework for using photogrammetry-based synthetic data generation to create an end-to-end deep learning pipeline for use in industrial applications.

While deep learning techniques have documented great success in many areas of computer vision, a key barrier that remains today with regard to large-scale industry adoption is the availability of data that can be used for model training. While large high-quality datasets are readily available for use with common object categories like animals and household items, this no longer holds in the case of many potential applications (e.g., the products in an industrial warehouse). For such applications, costly and labor-intensive data acquisition and labelling must first be carried out before deep

Corresponding author
Matthew Z. Wong, mzw17@ic.ac.uk

learning can be applied to the task at hand. As deep learning moves out of the academia and into industry, this limitation poses a serious problem for potential users: how can one cheaply and efficiently acquire training data when a large dataset does not already exist?

Working in collaboration with a leading UK online supermarket, we address the problem of dataset generation for fixed-appearance objects: object classes whose appearance have little to no change for all instances within a class. Such objects are ubiquitous in many potential deployment settings and include items such as consumer products, industrial goods, and machine parts, among others.

To that end, we propose combining the use of 3D modelling and render-based image synthesis to generate a synthetic dataset which can be used to train a deep convolutional neural network (CNN), a type of neural network which is frequently used for computer vision tasks. The inspiration comes from the following insight: for fixed (or low variation) appearance object classes, texture, and geometry information can be thought of as unchanging and can therefore be captured from a small number of physical samples without the risk of overfitting to individual specimens. Our approach builds on the existing literature by generating 3D scans of physical objects using photogrammetry. By rendering scenes from these realistic 3D models, we were able to generate a diverse array of synthetic images that were used as training data.

While previous work has made use of computer-generated 3D models to train CNNs, our study extends this concept by successfully demonstrating that this approach can be extended to 3D models acquired using photogrammetry.

In order to demonstrate our approach, we sought to train a deep learning model capable of performing the task of recognizing groceries. A total of 10 products were chosen for a classification task. Our results were promising: using our approach, we were able to successfully train and optimize a CNN that achieves a maximum classification accuracy of 95.8% on a general environment test set.

Furthermore, our use of synthetic training data generation has also enabled the automatic annotation and segmentation of training data. This has allowed us to train an object-detector for the same set of objects.

## RELATED WORK

### 3D Modelling in deep learning

3D Modelling has long been a mainstay of computer vision research. Nonetheless, it is only more recently that its potential applications to deep learning-based image classification and object detection have been considered, with 3D modelling used in conjunction with CNNs to train networks for use on *real* images. *Su et al. (2015)* demonstrate the use of 3D models for viewpoint estimation. By creating a database of millions of *rendered* training images using CAD models drawn from existing 3D model repositories, they were able to train a CNN that outperformed state-of-the-art methods on test sets containing *real* images. Similarly, *Peng et al. (2015)* and *Sarkar, Varanasi & Stricker (2017)* use a large number of 3D CAD models of objects to render realistic looking training images; the output was used to train a CNN model for classifying real world images of the objects.

*Tremblay et al. (2018)* extend this approach by applying synthetic data generated from 3D CAD models to the problem of object detection; they demonstrate that using domain randomization (i.e., varying parameters such as lighting, pose, and object textures), it is possible to train a compelling object detection network on automatically generated, non-photorealistic, synthetic data.

While previous work has exclusively made use of computer-generated 3D models (CAD software), our study extends this concept by successfully demonstrating that this approach can be extended to 3D models acquired using photogrammetry.

### 3D multi-image photogrammetry

3D multi-image photogrammetry is the process of reconstructing the three-dimensional properties of a selected object from a set of multiple two-dimensional images. Using such a system, one is able to accurately recreate the surface geometry, texture, color, and shape of the target object in a realistic manner.

The principles and techniques involved in multi-image photogrammetry have been the subject of much active computer vision research and have been comprehensively explored in the literature (*Faugeras, 1992*; *Linder, 2018*; *Luhmann et al., 2007*). The fundamental mathematical model of photogrammetry is central projection imaging (*Luhmann et al., 2007*). In this model, every 2D image is first used to generate a spatial bundle of rays; when all the ray bundles from multiple images are intersected, triangulation can be used to simultaneously orient all images and calculate every three-dimensional object point location.

State-of-the-art commercial systems are currently able to perform multi-image photogrammetry with minimal effort and without the need for calibrated cameras (*Agisoft LLC, 2018*). It is even possible to perform photogrammetry using only a smartphone camera (*EyeCue Vision Technologies, 2018*). Currently available photogrammetry systems thus give us the ability simply and effectively generate 3D models from real-world objects, thereby paving the way for our object-to-model deep learning paradigm.

## METHODS

Figure 1 shows the overall design of our object-to-model training pipeline, which is comprised of four main stages. The first two stages, 3D modelling and image rendering, are used to generate a synthetic dataset from physical samples of the target objects, while the next two stages are used to train and validate a deep learning model trained on the synthetic dataset.

- **3D Modelling:** This stage involves scanning physical products to produce 3D models. These include texture and color representations of the product and are of high enough quality to produce realistic images in the next stage.
- **Image rendering:** This stage produces a specified number of synthetic training images for each object which vary object pose, lighting, background, and occlusions.
- **Network training:** For the purposes of this study, several deep CNN architectures were trained to classify grocery items using rendered images generated from the image rendering step.

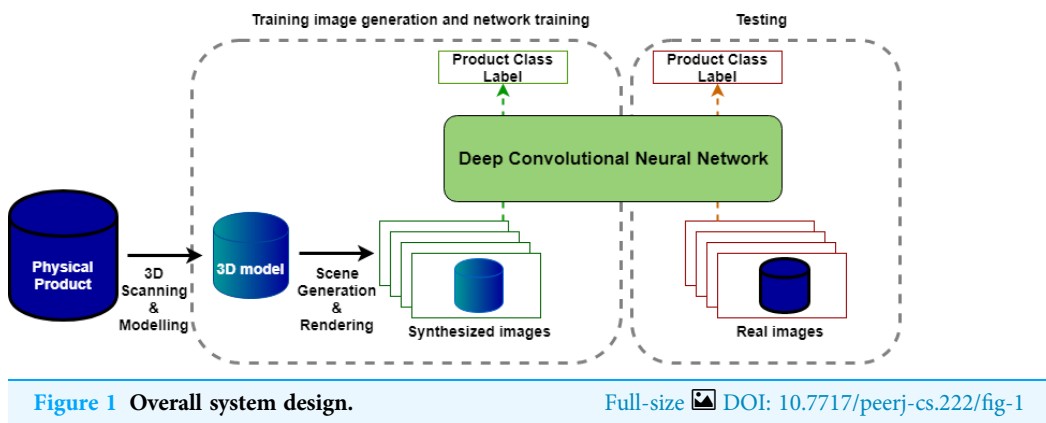

**Figure 1 Overall system design.**   

- **Testing and evaluation:** The trained classifier is then tested on a relatively small number (100 images per class) of test images collected.

These four stages fully define our end-to-end pipeline for generating and evaluating an image recognition network, for a number of specified fixed-appearance objects. In essence, the full pipeline takes in a set of physical objects and outputs a trained model. Each of the four stages will be described in detail in the sections below.

## 3D Modelling

The goal of the 3D modelling stage was to demonstrate a means by which 3D models of target objects can be feasibly created to be used as input for the image synthesis stage. The method employed required the constructed models to be of high enough quality to generate photo-realistic and consistent representations, as well as being economical to employ in a research setting.

Photogrammetry was used as a tool of choice. This technique takes 2D images of a 3D surface (photographs captured using a conventional camera) as input, and attempts to reconstruct the surfaces primarily using texture cues on the surfaces by following the steps below:

- **Camera calibration**: This is done automatically using matching features in the images, and estimating the most probable arrangement of cameras and features. A sparse point cloud of features on the modelled surface is calculated.
- **Mesh generation**: The point cloud is then triangulated and used to create a structural mesh of the surface.
- **Texture generation**: Texture information on the mesh surface is recovered by combining information from the original images.

Figure 2 shows a visualization of the steps described above as applied to an example object. This approach has allowed us to fully capture all geometry and texture information of all modelled objects. Minimal human effort is required—typically no more than 40 images were required per model, which can be done manually at a rate of approximately 5 min per product. In an industrial setting, this can be carried out even faster using

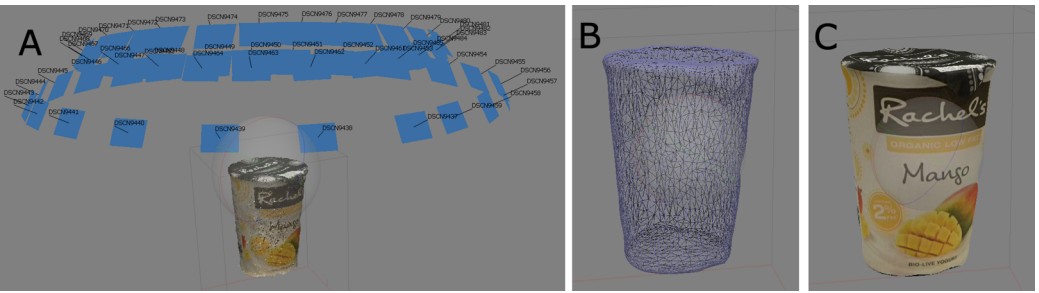

**Figure 2 Visualization of the steps in photogrammetry: (A) camera calibration and point cloud generation; (B) after mesh generation; (C) after texture generation.**

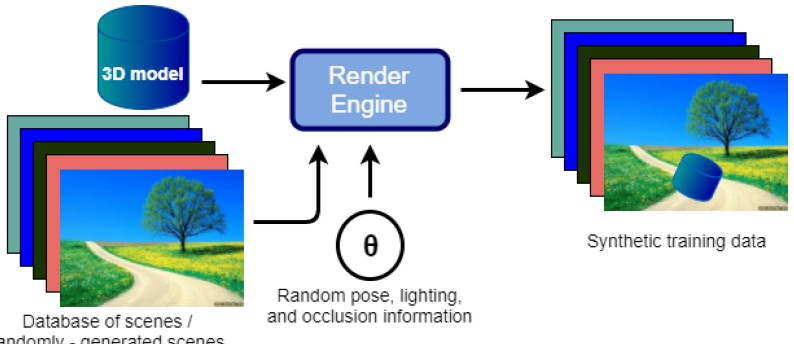

**Figure 3 Flow diagram for synthetic image rendering.**

various commercial photogrammetry-based photo-capture systems which can be used to automate the 3D modelling process.

## Image rendering

The goal of the image rendering stage is to produce an infinite supply of high-quality training data. This stage involves using a rendering engine to render a pose image of the 3D model. The appearance of this image depends on a user-defined distribution of rendering parameters $\theta$. This is combined with a background to generate the final training image. These steps are repeated to generate a potentially unlimited number of training images. Figure 3 provides an illustration of this process.

Scene appearance was fully defined with the following parameters: camera position w.r.t object (defined via an azimuth $\theta$ and elevation $\varphi$), camera distance to the object, lighting intensity (equivalently distance), number of lights.

As a simple heuristic, the camera location was defined to be evenly distributed around rings in a spherical coordinate system. This is illustrated in Fig. 4. The reason for this choice of distribution was due to the fact that it corresponded to common viewpoints of handheld grocery items.

Lamp locations were distributed evenly on a sphere. Additionally, camera-subject distance and lamp energies were distributed according to truncated normal distributions to

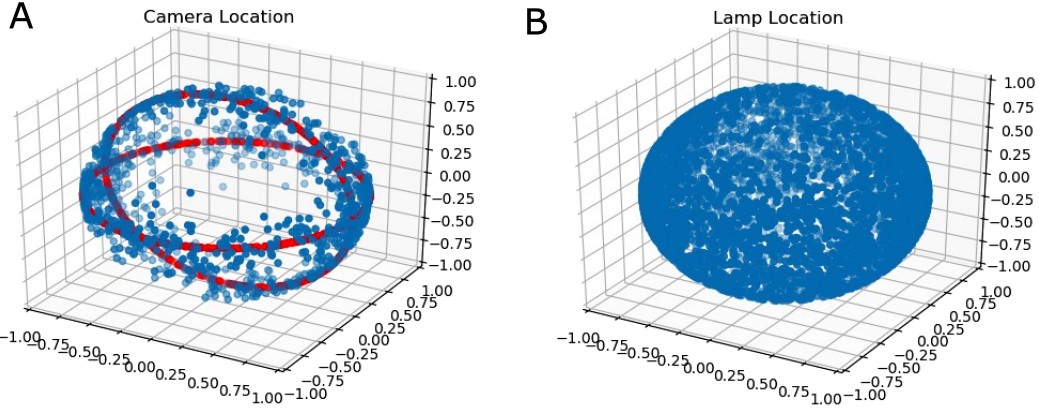

**Figure 4** (A) The rings on the shell (red rings) around which random normalized camera locations are sampled from (blue points); (B) The uniform distribution on the sphere of lamp locations.

ensure no negative energy lamps were generated. Point lamps were used primarily as a simple lighting model.

### Location distributions

We generated location distributions according to the method below. Assuming that one could define Cartesian coordinates $(x, y, z)$ in terms of spherical coordinates (defined using radius $\rho$, azimuth $\theta$, and elevation $\varphi$) as:

$$x = \rho \, \cos\theta \, \sin\phi \quad y = \rho \, \cos\theta \, \sin\phi \quad z = \rho \, \cos\phi \tag{1}$$

Distribution of camera locations were defined by stating:

$$\rho \sim \mathcal{T}(\mu_\rho, \sigma_\rho, a_\rho, b_\rho) \tag{2}$$

$$\phi \sim \mathcal{T}(0, \sigma_\phi, -\pi/2, \pi/2) \tag{3}$$

$$\theta \sim \mathcal{U}(0, 2\pi) \tag{4}$$

Where $\mathcal{T}(\mu, \sigma, a, b)$ is the truncated normal distribution, with mean $\mu$ and standard deviation $\sigma$, $a$, and $b$ define the limits of the distribution, for which the probability density function is zero outside the limits. Therefore if $X \sim \mathcal{T}(\mu, \sigma, a, b)$, then $X \sim \mathcal{N}(\mu, \sigma)$ if $a \leq X \leq b$. This set of variables defines a distribution around a ring in the $X$–$Y$ plane (with a normal $Z$), and the width of the ring can be controlled by specifying $\sigma_\phi$. If $\mathbf{x} = (x, y, z)$ is drawn from the distribution, one can "flip" the ring to have normal aligned with the $X$ axis by doing:

$$\mathbf{x}' = \mathbf{R_y}(\pi/2)\mathbf{x} \tag{5}$$

Where $\mathbf{R_y}(\pi/2)$ is the rotation matrix that rotates the point about the axis $y$, $\pi/2$ radians. This way points can be distributed around multiple rings (specified by their normals). For the purposes of this project, camera was distributed about the rings with the $Y$ and $Z$ axes as normals.

It is also worth mentioning that setting $\sigma_\phi$ to roughly $\pi/3$ radians generates a roughly uniform distribution of points around a sphere. This approach was used for generated lamp distributions.

### Lighting conditions

The number of point light sources $n_L$ was also sampled from a uniform distribution of non-negative integers (the min and max can be user-defined). The lamp energies $E$ were sampled using a truncated normal distribution:

$$n_L \sim \mathcal{U}\{n_{\min}, n_{\max}\} \tag{6}$$

$$E \sim \mathcal{T}(\mu_E, \sigma_E, 0, +\infty) \tag{7}$$

### Background scene generation

The rendered pose image is generated with a transparent background. Alpha composition was used to combine the rendered pose image with a background image. The resulting generated images are highly varied in terms of appearance. While a significant proportion of images seemed to look "unrealistic" (e.g., a yogurt pot in the International Space Station), the aim was not to achieve a simulation with perfect realism. Instead, the focus was on achieving enough background variation within the training data so that the final trained network would be as robust as possible, while ensuring that the *objects* themselves were rendered as realistically as possible.

Figure 5 shows examples of synthetic training data generated using the method described above. Note the variety of poses and lighting conditions represented, as well as the multitude of different backgrounds.

## Network training

The aim of the network training stage was to take a generated dataset as an input and produce a trained neural network that could be used to classify products from the dataset. The tool of choice for this task was a CNN, a class of neural networks that are frequently used for image classification tasks. CNNs are specialized neural networks that perform transformation functions (called convolutions) on image data. Deep CNNs contain hundreds of convolutions in series, arranged in various different architectures. It is common practice to take the output of the CNN and input it into a regular neural network (referred to as the fully-connected (FC) layer) in order to perform more specialized functions (in our context, a classification task). We refer readers to *Rawat & Wang (2017)* for a comprehensive review of the use of Deep CNNs in Image Classification.

## EXPERIMENTAL SETUP

### 3D Modelling

In practice, our 3D modelling stage involved two main steps:

- Step 1: Image capture of the objects to be modelled.
- Step 2: The generation of 3D models from captured images using a third party, free-for-research photogrammetry software.

## A-T: Examples of generated synthetic images

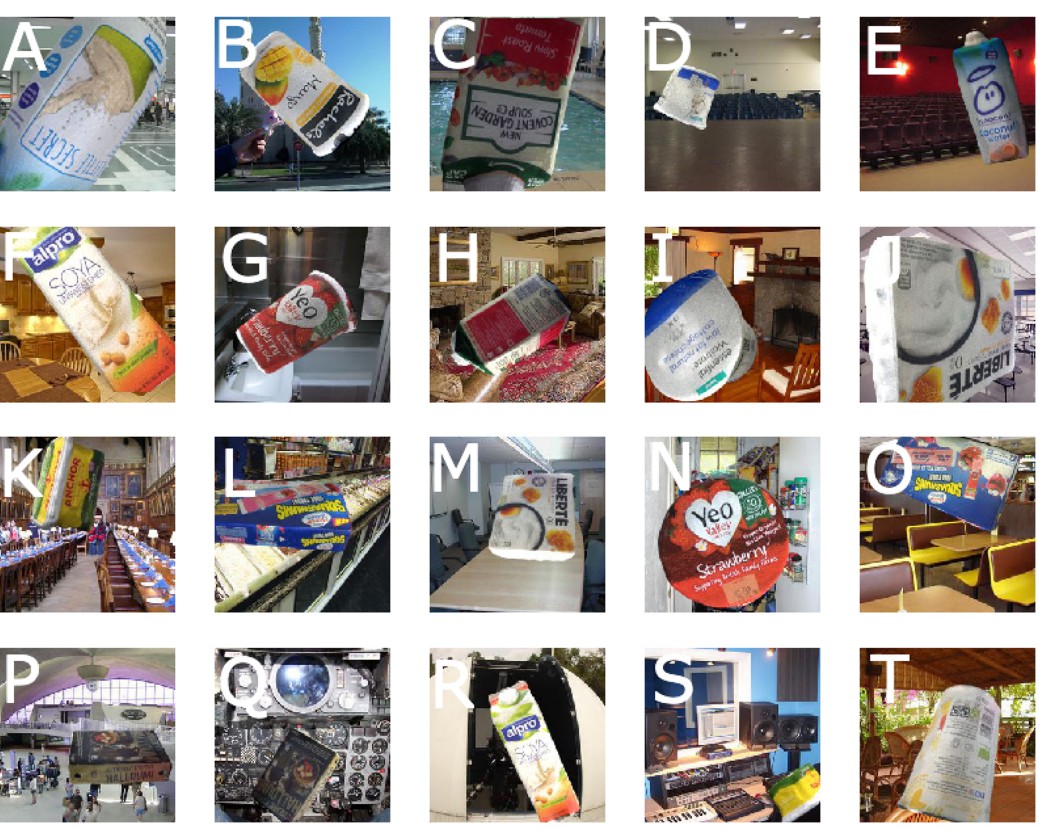

**Figure 5** Examples of generated synthetic images (A–T).

The first step involved capturing good images of the modelled object. The key here is to capture images so as to aid in the three steps for photogrammetry described in the 3D modelling subsection. Since camera calibration is performed by matching sets of features between groups of images to estimate the extrinsic (geometric) camera parameters, it is paramount we had good overlap between images and a large number of unique features to aid in reconstruction. Figure 6 shows an example subset of sample images captured for 3D modelling.

For most of our object models, which generally measured no more than 20 cm at the longest, the distance for image capture was set at about 40 cm away from the object. Images were captured at two levels, roughly 30° and 60° elevation from the center of volume of the object. The azimuth of the camera about the object was changed with increments of 20° (see Fig. 2 for a visualization of typical camera placement relative to the object). A base with distinct patterns was also placed below the captured object to aid in feature matching.

In the second step of the 3D modelling stage, AgiSoft Photoscan® was used with the images from the previous step as input. This is free-for-research software and can be used by requesting a free license (https://www.agisoft.com/buy/licensingoptions/). The software offers an automated pipeline that performs camera calibration/alignment and point cloud

A–D: Example images captured for modelling yogurt pot

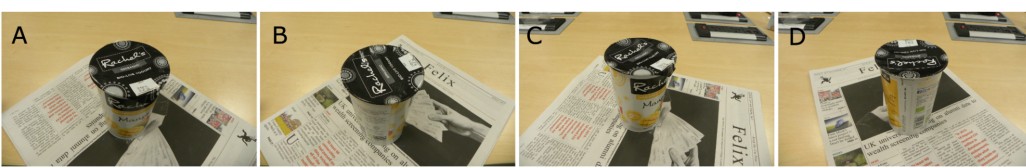

**Figure 6** **Example subset of images captured for the modelling of a yogurt pot at different elevations and angles (A–D).** Notice how a magazine was used as a base to create more keypoints for camera alignment.

**Table 1 The runtimes for rendering.**

**Rendering (100K images)**

| Samples | Resolution (px) | Runtime (h) |
|---|---|---|
| 64 | 224 | 4 |
| 128 | 224 | 6 |
| 64 | 300 | 5.5 |
| 128 | 300 | 9 |

Note:
Rendering runtimes depend heavily on rendering settings. Hardware used was an Nvidia GTX Titan X GPU.

generation simultaneously. Mesh generation is then followed by texture generation. This generates a mesh file in the .obj format, which contains positions of vertices in space, its UV coordinate and polygons in the form of vertex lists, and a texture .jpeg file.

## Image rendering

Blender, an open-source software, was used for the rendering of the synthetic training data. A Python layer was built on top of the Blender API https://docs.blender.org/api/current/index.html allowing for programmatic control of rendering parameters, which implemented the rendering algorithm described in the Methods section, and enabled automated generation of training data at scale. The output of this layer was a $300 \times 300$ pixels RGB alpha image, which contained the product of interest in random orientation and random lightning, while keeping the alpha values of the remaining pixels as zero. The final image was created by merging it with a random background image from the SUN database of over 80,000 images (*Xiao et al., 2010*). For this purpose a Python wrapper around the open-source Python Imaging Library was built. Both Blender and the PIL wrappers have been open sourced with this paper to facilitate reproducibility and future work (https://github.com/921kiyo/3d-dl).

## Datasets

### Training data

Using our 3D modelling and image rendering methods, we generated a training data set which consists of 100,000 training images for 10 different grocery products[1] (10,000 images per class) with manually chosen values for the distributions of rendering parameters. Figure 5 shows examples of the generated training data.

[1] Ten grocery products used in our study are 1. Anchor Spreadable, 2. Innocent Coconut Water, 3. Essential Waitrose Low Fat Natural Cottage Cheese, 4. Liberte Honey Greek Style Yogurt, 5. Rachel's Organic Greek Style Lemon, 6. Yeo Valley Organic Strawberry Yogurt, 7. New Covent Garden Slow Roast Tomato Soup, 8. Alpro Longlife Original Soya Milk Alternative, 9. Munch Bunch Strawberry Squashums, 10. Cypressa Traditional Halloumi Cheese.

## A-H: Example images from proprietary general environment test set

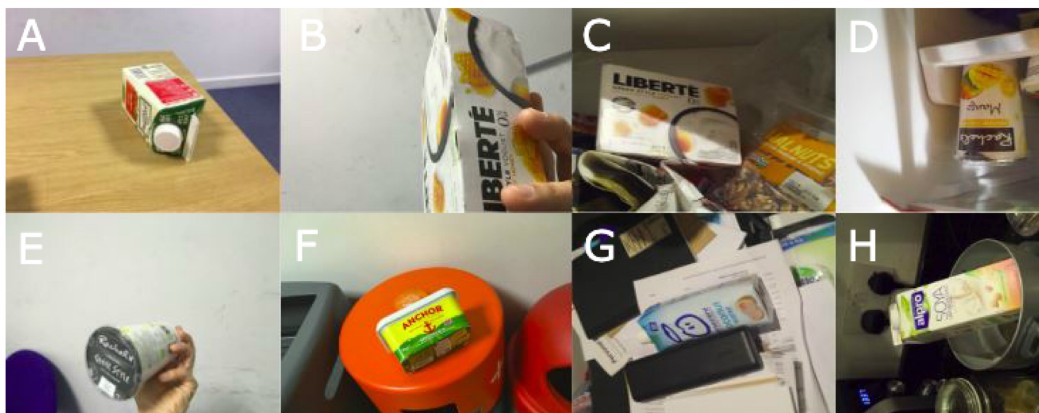

**Figure 7 Sample of proprietary general environment test set (A–H).**

The durations for generating training images is shown in Table 1. The training set used in our experiments took approximately 9 hours to generate using our NVIDIA GTX TITAN X GPU.

### Test data

To evaluate the classifier's generalization performance, a special test and validation set was acquired by conventional image capture. As the aim of our method was to create training data for unique fixed-appearance objects for which no suitable dataset already exists, it was not considered constructive to make use of standard test sets such as ImageNet *Deng et al. (2009)*, given that these test sets generally focus on generic object categories (e.g., car) rather than specific individualized objects. Figure 7 displays some examples of our test set which was manually acquired in a variety of locations. The set contains 1,000 images of 10 classes (100 images per class) from a variety of perspectives, at different distances, in different lighting conditions and with and without occlusion. These images were acquired with a number of different devices, including smartphones, DSLR cameras, and digital cameras.

### Network training

For network training, three different CNN architectures were tested using the Keras API (*Chollet, 2015*). These were Google's InceptionV3 (*Szegedy et al., 2014*), the Residual Network (ResNet-50) (*He et al., 2015*) architecture, and the VGG-16 (*Simonyan & Zisserman, 2014*) architecture.

A FC layer with 1,024 hidden nodes and 10 output nodes for each class was defined, and a standard stochastic gradient descent optimizer with momentum was used. For all of the above architectures, the only parameter that was manipulated was the number of trained convolutional layers (the FC layers are always trained), ranging from zero layers to all layers. The final reported models were trained with all convolutional layers unfrozen for retraining. The weights for the convolutional layers were initialized based on those

| Table 2 Constant variables for network training. | |
|---|---|
| Learning rate (InceptionV3) | 0.00256 |
| Learning rate (VGG16, ResNet50) | 0.0001 |
| Image input size (px, px) | (224,224) |
| Batch size | 64 |
| Number of fully-connected layers | 2 |
| Hidden layer size | 1,024 |
| Optimizer | SGD |
| Epochs | 12 |

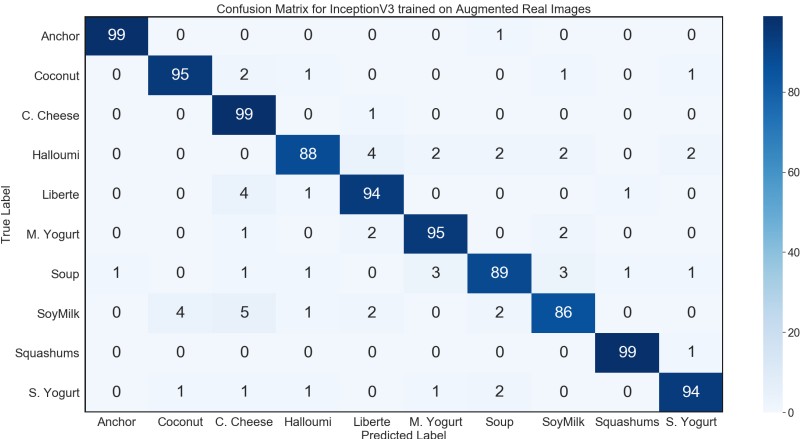

**Figure 8 Confusion matrix for InceptionV3 using synthetic data on general environment test set.**

trained using the ImageNet dataset (*Deng et al., 2009*). All other parameters were kept constant as shown in Table 2. Additionally, for the InceptionV3 model a grid-search was carried out to fine-tune the learning rate used.

All computation was performed on a system using an Intel Xeon E5-1630 v3 3.70 GHz CPU with 31 GB memory and an NVIDIA GTX TITAN X GPU. A virtual environment was used with Tensorflow 1.4.1 and Keras 2.1.3 installed.

# EXPERIMENTAL RESULTS

We evaluated the performance of our system on two common computer vision tasks, namely image classification and object detection.

## Image classification

The first task evaluated was to perform image classification on the images in our general environment test set. We were able to achieve a test accuracy of 95.8% using the InceptionV3 architecture with the optimized learning rate and all convolutional layers being retrained. The confusion matrix for this network is shown in Fig. 8.

**Table 3 Summary of testing results (accuracy, data source, average precision, average recall) for different architectures.**

| Network | Data source | Accuracy | A. Precision | A. Recall |
|---|---|---|---|---|
| InceptionV3 | Synthetic images | 95.8 | 96.0 | 95.8 |
| ResNet50 | Synthetic images | 93.8 | 93.9 | 93.8 |
| VGG16 | Synthetic images | 83.2 | 84.6 | 83.2 |
| InceptionV3 | Augmented real images | 64.5 | 75.7 | 64.5 |

The trained network was able to classify almost every single image that it could reasonably have been expected to successfully identify. In fact, the majority of the few misclassified images showed the underside of products. From this point of view, the product is usually seen as a white circle or rectangle. This means that there are few to none distinct features by which the different products can be distinguished, making the classification of products shown from the underside extremely challenging, even for humans.

Experimentation with different CNN architectures yielded favorable results, with the InceptionV3 architecture achieving the best performance. These results are shown in Table 3.

These results demonstrate that a classifier can be successfully trained using only synthetic data acquired using photogrammetry techniques. To our knowledge this is the first demonstration of the use of photogrammetry-based synthetic data in successfully training a CNN.

### Benchmark experiment

To further demonstrate the value of our approach, we also performed a comparison experiment evaluating the performance of a network trained on real images, as opposed to the synthetic data used in our proposed method. To ensure a meaningful comparison, the real images used were the same images used to create the 3D models for synthetic data generation.

During the 3D modelling stage, we used a DSLR camera to capture 60 real images of each product from all angles, and subsequently used about 30–40 of those images to generate the 3D models for use in synthetic data. For this control experiment, we used transfer learning to train a network on those 60 images per class, using a data generator with image augmentation to produce an arbitrarily large dataset for training. To carry out transfer learning, we initialized an InceptionV3 network with ImageNet weights, further trained all layers using the augmented real images. InceptionV3 with transfer learning achieved convergence. However, this performance failed to generalize adequately, with the trained model achieving only 64.5% accuracy on the general environment test set; while demonstrating some evidence of learning, this result fell far below the model trained on synthetic images. The confusion matrix for this network is shown in Fig. 9.

The results of the control experiment show that in situations where only a small number of training images can be captured, our method of generating synthetic training data from 3D models is more effective than the alternative of using image augmentation.

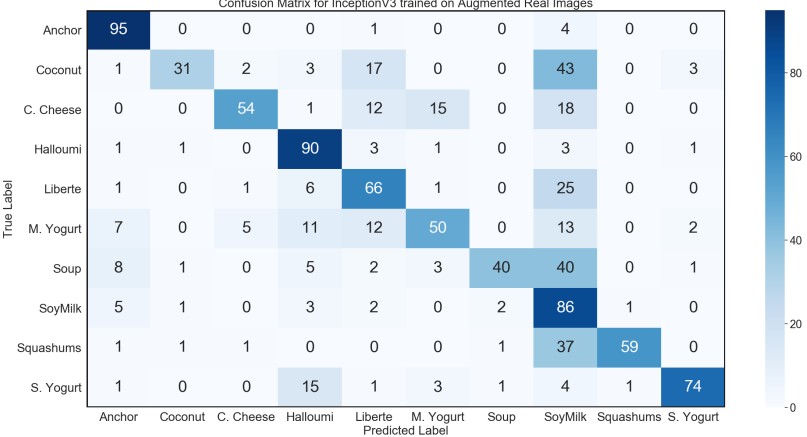

**Figure 9 Confusion matrix for InceptionV3 benchmark experiment on general environment test set.**

## Object detection

Our approach opens up the opportunity to train object detectors and segmentation algorithms given that our pipeline can produce pixel-level annotations of products. This is possible as the placement of the object of interest in the image is fully under our control, and can be logged automatically. For this application, we have logged the object bounding box. This information can then be compared with the detector estimated bounding box. This provides a huge advantage as no manual pixel annotation is necessary as is the case with currently available datasets, e.g., the Microsoft Coco dataset (*Lin et al., 2014*). First experiments using RCNNs on our dataset show strong results when performing object detection on our general environment test set.

The chosen architecture for this task is the RetinaNet architecture (*Lin et al., 2017*). This is a one-stage architecture that runs detection and classification over a dense sampling of possible sub-regions (or "Anchors") of an image. The authors have claimed that it outperforms most state-of-the-art one-stage detectors in terms of accuracy, and runs faster than two-stage detectors (such as the Fast-RCNN architecture (*Girshick, 2015*)). A "detection as classification" (DAC) accuracy metric was calculated by using the following formula:

$$\text{DAC} = \frac{\sum_i^n \text{TP}_i}{n} \tag{8}$$

Where the quantity $\text{TP}_i$ is summed over every image $i$, and is calculated:

$$\text{TP}_i = \begin{cases} 1 & \text{if top 3 detections for i contain correct class} \\ 0 & \text{otherwise} \end{cases} \tag{9}$$

The same general environment test set introduced above was used to test the accuracy of this learning task, when the same set of rendered training images were used. The rendered images were split into a train and validation set. The validation split of the rendered images is used to measure the deviation in network performance between real

## A-J: Examples of detection results on test images

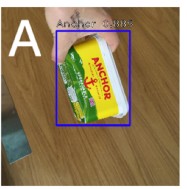 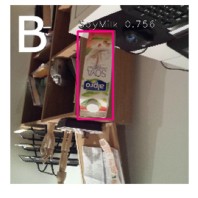 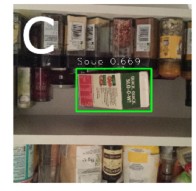 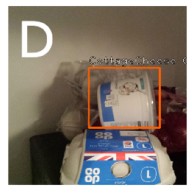 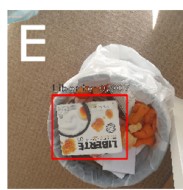

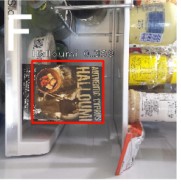 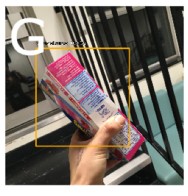 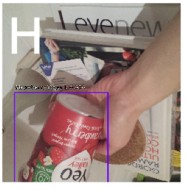 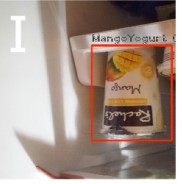 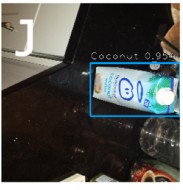

**Figure 10 Detection bounding boxes and confidence scores for classification (A–J).**

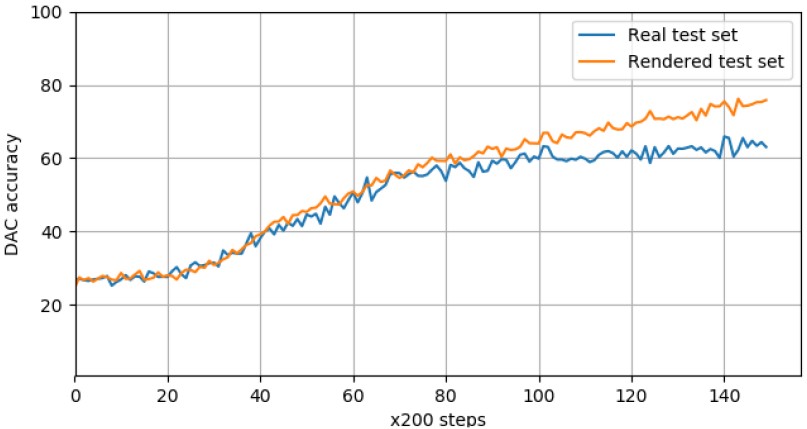

**Figure 11 Real and rendered image validation scores for product detection, logged every 200 training steps.** The DAC accuracy for real images deviates from the rendered accuracy at around 20K steps.

and rendered images. The number of rendered validation images per class used was the same as that of the real test images. Figure 10 highlights the performance of the detection algorithm when applied to a set of sample test images.

The reported DAC accuracy for the test set was 64%. Upon closer inspection of the test results, it was discovered that the detection bounding boxes were mostly accurately calculated. However, the main factor driving the score down was incorrect classifications. This classification loss was less pronounced in the rendered image validation dataset, giving a final accuracy of 75%.

The logged validation accuracy for both real and rendered data is shown in Fig. 11, and indicates a deviation between how the network perceives real images of objects, and rendered images. This is a surprising finding, given near-perfect performance on the classification task as noted above. This shows that our current set of rendering parameters might not be as robust as previously thought, and can be dependent on the learning task.

It is likely that further optimization of the rendering pipeline can be done in order to optimize against the detection task, and to make it more robust against a larger variety of learning tasks.

## DISCUSSION

### Novel contribution

In the course of our study, we developed an original pipeline that goes from data acquisition, over data generation to training a CNN classifier. This is a novel combination of several tools including photogrammetry for 3D scanning and the use of a graphics engine for training data generation. The 95.8% accuracy achieved on the general environment test set demonstrates that photogrammetry-based 3D models can be successfully used to train an accurate classifier for real-world images. To our knowledge this the first-time photogrammetry-based 3D models were used to train a CNN. In addition, we developed an end-to-end software tool for synthetic dataset generation, which includes the development of Blender API, Scene generation library as well as the evaluation of the classifier. All the code used in our study has been open-sourced (https://github.com/921kiyo/3d-dl).

### Limitations

Throughout the research conducted into this approach, two main limitations became apparent.

First, using photogrammetry for 3D modelling results in reconstruction noise in the case of transparent features and large unicolor areas, thereby reducing overall network performance. This could be mitigated by combining photogrammetry with other information sources. For example, products from *Occipital, Inc. (2018)* use an infrared iPad mount to combine information from a typical camera with information from an infrared sensor to create more accurate 3D models. Industrial-grade RGBD scanners which combine depth and color information are also readily available on the market, allowing for the creation of 3D models of far greater accuracy.

Second, the process of acquiring 3D models and preparing them for rendering required roughly 15 min of manual work per product. While this is manageable for a small product set, this would become an increasingly serious limitation as the number of products to be scanned increases. A deployment-ready solution using our method will require a more sophisticated approach to acquiring the product photos in order to minimize the amount of manual labor required. This could include investments into appropriate hardware such as the above-mentioned industrial grade 3D scanners as well as automating the scanning process. For larger products (e.g., cars) a large room or warehouse could be fitted with multiple movable cameras to capture photographs for photogrammetry.

## FURTHER WORK

The viability of using photogrammetry-based data synthesis for training deep neural networks sets the stage for further research capitalizing on the advantages of this approach.

For example, an area holding great promise is that of *expanded hyperparameter optimization*. Standard neural network pipelines generally allow for the optimization of network hyperparameters, whilst assuming a fixed pre-existing dataset (*Bergstra & Bengio, 2012*). The system described here, which performs both data generation and model training, can be thought of as a black box with tunable hyperparameters that include both rendering and network training hyperparameters. This expanded set of hyperparameters provides a more expressive and robust model to maximize performance on real-world computer vision tasks. For example, consider the task of detecting objects in a low-light environment. By measuring performance on real validation images taken from this environment, an appropriate optimization procedure (e.g., a greedy sequential search, or Bayesian optimization (*Snoek, Larochelle & Adams, 2012*; *Bergstra et al., 2011*)) may be able to efficiently find a suitable lighting distribution in our rendering framework which maximizes performance of the model by optimizing network training, as well as optimizing the training data itself.

## CONCLUSION

The insatiable need for ever-larger quantities of quality training data is one of the main limitations of deep learning. As deep learning is applied more extensively, this limitation will become increasingly evident. Novel applications of deep learning will require novel datasets which will need to be manually created with great cost and effort. In this paper, we have shown that by generating synthetic training data using photogrammetry, we are able to produce training data of sufficiently high-quality for use in deep learning applications, while significantly reducing the amount of work and cost associated with data collection. Furthermore, the image generation process allows automatic pixel annotation which allows for the training of detection models. We believe our work holds particular promise for use in industrial applications, where recognition and detection tasks tend to involve a large range of unique objects for which there is likely not to be pre-existing datasets. Our method thus opens up new opportunities for the application of deep learning to fields where large-scale data collection and dataset curation has previously been unfeasible or expensive.

## ACKNOWLEDGEMENTS

We would like to thank Dr Bernhard Kainz at Imperial College London for his advice and valuable feedback, and to Ocado Group, particularly Luka Milic and David Sharp in the 10x Technology Team, for insightful discussions and suggestions.

### Funding

The authors received no funding for this work.

### Competing Interests

The authors declare that they have no competing interests.

## Author Contributions

- Matthew Z. Wong conceived and designed the experiments, performed the experiments, analyzed the data, contributed reagents/materials/analysis tools, prepared figures and/or tables, performed the computation work, authored or reviewed drafts of the paper, approved the final draft.
- Kiyohito Kunii conceived and designed the experiments, performed the experiments, analyzed the data, contributed reagents/materials/analysis tools, prepared figures and/or tables, performed the computation work, authored or reviewed drafts of the paper, approved the final draft.
- Max Baylis conceived and designed the experiments, performed the experiments, analyzed the data, contributed reagents/materials/analysis tools, prepared figures and/or tables, performed the computation work, authored or reviewed drafts of the paper.
- Wai Hong Ong conceived and designed the experiments, performed the experiments, analyzed the data, contributed reagents/materials/analysis tools, prepared figures and/or tables, performed the computation work, authored or reviewed drafts of the paper, approved the final draft.
- Pavel Kroupa conceived and designed the experiments, performed the experiments, analyzed the data, prepared figures and/or tables, performed the computation work.
- Swen Koller conceived and designed the experiments, performed the experiments, analyzed the data, prepared figures and/or tables, performed the computation work, authored or reviewed drafts of the paper.

## Data Availability

Code is available at GitHub: https://github.com/921kiyo/3d-dl.

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
