# Peer review of "Synthetic dataset generation for object-to-model deep learning in industrial applications"

_PeerJ Computer Science, doi:10.7717/peerj-cs.222_

## Round 0.1 · original submission · Major Revisions

Both of the reviewers have pointed out some critical issues that should be addressed. The authors are suggested to make revisions accordingly.

Reviewer 1 ·

Basic reporting

The authors propose a method based on 3D modeling and photogrammetry to create synthetic datasets of objects for warehouse management. The results confirm the validity of the approach, in the form of a CNN trained on the synthetic data and achieving high classification results in real world applications.
The paper is interesting, technically sound, and well-written. However, some details are too sparsely described, such as:
• The ‘3d modeling’ section is rather sparsely described. Maybe consider extending the description to help ensure reproducibility of the procedure.
• More details on the synthetic datasets and the training data should be provided. How many classes are generated? How many images per class? What is the size of the image?
• More details on the training procedure should also be provided. E.g., how many epochs are used to train the network? On which computing platform?
Based on these considerations, it is my opinion that the paper requires a major revision to increase the clarity of the description and ensure the reproducibility of the procedure and the results.

Experimental design

Experimental procedure is too sparsely described. In particular:
• More details on the synthetic datasets and the training data should be provided. How many classes are generated? How many images per class? What is the size of the image?
• More details on the training procedure should also be provided. E.g., how many epochs are used to train the network? On which computing platform?

Validity of the findings

'no comment'

Reviewer 2 ·

Basic reporting

1.Some figures are fuzzy. For example, the Figure 1 is a little bit blurry, it could be better if a higher resolution one is used. Besides, in the Figure 7, the label of abscissa is incomplete.

Experimental design

1. In the subsection of Image Classification, authors just use their synthetic dataset to do the experimentation with different CNN architectures. However, they also need to use real datasets for comparative experiments to make their results convincing.

Validity of the findings

1. The whole paper lacks enough innovation points, and the methods used are mostly those proposed by others. The author needs to expound his new method and innovation point emphatically

Additional comments

In the section of LIMITATIONS, the paper shows that "the process of acquiring 3D models and preparing them for rendering required roughly 15 minutes of manual work per product." When the object is large, the author should have further thoughts on how to efficiently scan it with less resources and time.

---

## Round 0.2 · accepted · Accept

Both reviewers and I believe that this revised version is acceptable for publication. We appreciate the careful revision by the authors.

Reviewer 1 ·

Basic reporting

It is my opinion that the authors answered the issues raised by the reviewers in a satisfactory way. Therefore, I have no further comments.

Experimental design

no comment

Validity of the findings

no comment

Reviewer 2 ·

Basic reporting

no comment

Experimental design

no comment

Validity of the findings

no comment

Additional comments

Authors have well corrected the deficiencies and weaks I put forward.I'm satisfied with the revised paper.